# SVMAX: A FEATURE EMBEDDING REGULARIZER

## ABSTRACT

A neural network regularizer (*e.g.*, weight decay) boosts performance by *explicitly* penalizing the complexity of a network. In this paper, we penalize inferior network activations – feature embeddings – which in turn regularize the network's weights *implicitly*. We propose singular value maximization (SVMax) to learn a uniform feature embedding. The SVMax regularizer integrates seamlessly with both supervised and unsupervised learning. During training, our formulation mitigates model collapse and enables larger learning rates. Thus, our formulation converges in fewer epochs, which reduces the training computational cost. We evaluate the SVMax regularizer using both retrieval and generative adversarial networks. We leverage a synthetic mixture of Gaussians dataset to evaluate SVMax in an unsupervised setting. For retrieval networks, SVMax achieves significant improvement margins across various ranking losses.

## 1 INTRODUCTION

A neural network's knowledge is embodied in *both* its weights and activations. This difference manifests in how network pruning and knowledge distillation tackle the model compression problem. While pruning literature Li et al. (2016); Luo et al. (2017); Yu et al. (2018) compresses models by removing less significant weights, knowledge distillation Hinton et al. (2015) reduces computational complexity by matching a cumbersome network's last layer activations (logits). This perspective, of weight-knowledge versus activation-knowledge, emphasizes how neural network literature is dominated by explicit weight regularizers. In contrast, this paper leverages singular value decomposition (SVD) to regularize a network through its last layer activations – its feature embedding.

Our formulation is inspired by principal component analysis (PCA). Given a set of points and their covariance, PCA yields the set of orthogonal eigenvectors sorted by their eigenvalues. The principal component (first eigenvector) is the axis with the highest variation (largest eigenvalue) as shown in Figure 1c. The eigenvalues from PCA, and similarly the singular values from SVD, provide insights about the embedding space structure. As such, by regularizing the singular values, we reshape the feature embedding.

The main contribution of this paper is to leverage the singular value decomposition of a network's activations to regularize the embedding space. We achieve this objective through singular value maximization (SVMax). The SVMax regularizer is oblivious to both the input-class (labels) and the sampling strategy. Thus it promotes a uniform embedding space in both supervised and unsupervised learning. Furthermore, we present a mathematical analysis of the mean singular value's lower and upper bounds. This analysis makes tuning the SVMax's balancing-hyperparameter easier, when the feature embedding is normalized to the unit circle.

The SVMax regularizer promotes a uniform embedding space. During training, SVMax speeds up convergence by enabling large learning rates. The SVMax regularizer integrates seamlessly with various ranking losses. We apply the SVMax regularizer to the last feature embedding layer, but the same formulation can be applied to intermediate layers. The SVMax regularizer mitigates model collapse in both retrieval networks and generative adversarial networks (GANs) Goodfellow et al. (2014); Srivastava et al. (2017); Metz et al. (2017). Furthermore, the SVMax regularizer is useful when training unsupervised feature embedding networks with a contrastive loss (*e.g.*, CPC) Noroozi et al. (2017); Oord et al. (2018); He et al. (2019); Tian et al. (2019).

In summary, we propose singular value maximization to regularize the feature embedding. In addition, we present a mathematical analysis of the mean singular value's lower and upper bounds

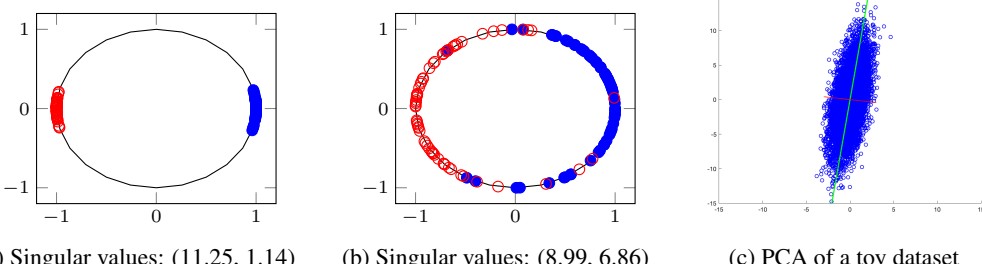

(a) Singular values: (11.25, 1.14)     (b) Singular values: (8.99, 6.86)     (c) PCA of a toy dataset

Figure 1: Feature embeddings scattered over the $2D$ unit circle. In (a), the features are polarized across a single axis; the singular value of the principal (horizontal) axis is large while singular value of the secondary (vertical) axis is small, respectively. In (b), the features are spread uniformly across both dimensions; both singular values are comparably large. (c) depicts the PCA analysis of a toy $2D$ Gaussian dataset to demonstrate our intuition. The principal component (green) has the highest eigenvalue, *i.e.*, the axis with the highest variation, while the second component (red) has a smaller eigenvalue. Maximizing all eigenvalues promotes data dispersion across all dimensions. In this paper, we maximize the mean singular value to regularize the feature embedding and avoid a model collapse.

to reduce hyperparameter tuning (Sec. 3). We quantitatively evaluate how the SVMax regularizer significantly boosts the performance of ranking losses (Sec. 4.1). And we provide a qualitative evaluation of using SVMax in the unsupervised learning setting via GAN training (Sec. 4.2).

## 2 RELATED WORK

Network weight regularizers dominate the deep learning regularizer literature, because they support a large spectrum of tasks and architectures. Singular value decomposition (SVD) has been applied as a weight regularizer in several recent works Zhang et al. (2018); Sedghi et al. (2018); Guo & Ye (2019). Zhang et al. (2018) employ SVD to avoid vanishing and exploding gradients in recurrent neural networks. Similarly, Guo & Ye (2019) bound the singular values of the convolutional layer around 1 to preserve the layer's input and output norms. A bounded output norm mitigates the exploding/vanishing gradient problem. Weight regularizers share the common limitation that they do not enforce an explicit feature embedding objective and are thus ineffective against model collapse.

Feature embedding regularizers have also been extensively studied, especially for classification networks Rippel et al. (2015); Wen et al. (2016); He et al. (2018); Hoffman et al. (2019); Taha et al. (2020). These regularizers aim to maximize class margins, class compactness, or both simultaneously. For instance, Wen et al. (2016) propose center loss to explicitly learn class representatives and thus promote class compactness. In classification tasks, test samples are assumed to lie within the same classes of the training set, *i.e.*, closed-set identification. However, retrieval tasks, such as product re-identification, assume an open-set setting. Because of this, a retrieval network regularizer should aim to spread features across many dimensions to fully utilize the expressive power of the embedding space.

Recent literature Sablayrolles et al. (2018); Zhang et al. (2017) has recognized the importance of a spread-out feature embedding. However, this literature is tailored to triplet loss and therefore assumes a particular sampling procedure. In this paper, we leverage SVD as a regularizer because it is simple, differentiable Ionescu et al. (2015), and class oblivious. SVD has been used to promote *low* rank models to learn compact intermediate layer representations Kliegl et al. (2017); Sanyal et al. (2019). This helps compress the network and speed up matrix multiplications on embedded devices (iPhone and Raspberry Pi). In contrast, we regularize the embedding space through a *high* rank objective. By maximizing the mean singular value, we promote a higher rank representation – a spread-out embedding.

## 3 SINGULAR VALUE MAXIMIZATION (SVMAX)

We first introduce our mathematical notation. Let $\mathcal{I}$ denote the image space and $E_{\mathcal{I}} \in R^d$ denote the feature embeddings space, where $d$ is the dimension of the features. A feature embedding network is a function $F_\theta : \mathcal{I} \rightarrow E_{\mathcal{I}}$, parameterized by the network's weights $\theta$. We quantify similarity between an image pair $(\mathcal{I}_1, \mathcal{I}_2)$ via the Euclidean distance in feature space, *i.e.*, $\|E_{\mathcal{I}_1} - E_{\mathcal{I}_2}\|_2$.

During training, a $2D$ matrix $E \in R^{b \times d}$ stores $b$ samples' embeddings, where $b$ is the mini-batch size. Assuming $b \geq d$, the singular value decomposition (SVD) of $E$ provides the singular values $S = [s_1, ., s_i, ., s_d]$, where $s_1$ and $s_d$ are the largest and smallest singular values, respectively. We maximize the mean singular value, $s_\mu = \frac{1}{d} \sum_{i=1}^{d} s_i$, to regularize the network's last layer activations – the feature embedding. By maximizing the mean singular value, the deep network spreads out its embeddings. This has the added benefit of implicitly regularizing the network's weights $\theta$. The proposed SVMax regularizer integrates with both supervised and unsupervised feature embedding networks as follows

$$L_{\text{NN}} = L_r - \lambda \frac{1}{d} \sum_{i=1}^{d} s_i = L_r - \lambda s_\mu, \tag{1}$$

where $L_r$ is the original network loss and $\lambda$ is a balancing hyperparameter.

**Lower and Upper Bounds of the Mean Singular Value:** One caveat to equation 1 is the hyperparameter $\lambda$. It is difficult to tune because the mean singular value $s_\mu$ depends on the range of values inside $E$ and its dimensions $(b, d)$. Thus, changing the batch size or embedding dimension requires a different $\lambda$. To address this, we utilize a common assumption in metric learning – the unit circle (L2-normalized) embedding assumption. This assumption provides *both* lower and upper bounds on ranking losses. This will allow us to impose lower and upper bounds on $s_\mu$.

For an L2-normalized embedding $E$, the largest singular value $s_1$ is maximum when the matrix-rank of $E$ equals one, *i.e.*, $rank(E) = 1$, and $s_i = 0$ for $i \in [2, d]$. Horn & Johnson (1991) provide an upper bound on this largest singular value $s_1$ as $s^*(E) \leq \sqrt{||E||_1 ||E||_\infty}$. This holds in equality for all L2-normalized $E \in R^{b \times d}$ with $rank(E) = 1$. For an L2-normalized matrix $E$ with $||E||_1 = b$, and $||E||_\infty = 1$, this gives:

$$s^*(E) = \sqrt{||E||_1 ||E||_\infty} = \sqrt{b}. \tag{2}$$

Thus, the lower bound $L$ on $s_\mu$ is $L = \frac{s^*(E)}{d} = \frac{\sqrt{b}}{d}$.

Similarly, an upper bound is defined on the sum of the singular values Turkmen & Civciv (2007); Kong et al. (2018); Friedland & Lim (2016). This summation is formally known as the nuclear norm of a matrix $||E||_*$. Hu (2015) established an upper bound on this summation using the Frobenius Norm $||E||_F$ as follows

$$||E||_* \leq \sqrt{\frac{b \times d}{max(b, d)}} ||E||_F, \tag{3}$$

where $||E||_F = \left( \sum_{i=1}^{rows} \sum_{j=1}^{cols} |E_{ij}|^2 \right)^{\frac{1}{2}} = \sqrt{b}$ because of the L2-normalization assumption.

Accordingly, the lower and upper bounds of $s_\mu$ are $[L, U] = [\frac{s^*(E)}{d}, \frac{||E||_*}{d}]$. With these bounds, we rewrite our final loss function as follows

$$L_{\text{NN}} = L_r + \lambda \exp\left( \frac{U - s_\mu}{U - L} \right). \tag{4}$$

The SVMax regularizer grows exponentially $\in [1, e]$. We employ this loss function in all our retrieval experiments. It is important to note that the L2-normalized assumption makes $\lambda$ tuning easier, but it is not required. Equation 4 makes the hyperparameter $\lambda$ only dependent on the range of $L_r$ which is also bounded for ranking losses.

**Lower and Upper Bounds of Ranking Losses:** We briefly show that ranking losses are bounded when assuming an L2-normalized embedding. Equations 5 and 6 show triplet and contrastive losses,

Table 1: Quantitative evaluation on CUB-200-2011 with batch size $b = 144$, embedding dimension $d = 128$ and multiple learning rates $lr = \{0.01, 0.001, 0.0001\}$. $\triangle_{R@1}$ column indicates the R@1 improvement margin relative to the vanilla ranking loss. A large learning rate $lr$ increases the chance of model collapse, while a small $lr$ slows convergence. $\lambda$ is dependent on the ranking loss.

| | $lr = 0.01$ | | | | $lr = 0.001$ | | | | $lr = 0.0001$ | | | |
|---|---|---|---|---|---|---|---|---|---|---|---|---|
| Method | NMI | R@1 | R@8 | $\triangle_{R@1}$ | NMI | R@1 | R@8 | $\triangle_{R@1}$ | NMI | R@1 | R@8 | $\triangle_{R@1}$ |
| | | | | | | Contrastive | | | | | | |
| Vanilla | 0.435 | 25.73 | 58.88 | - | 0.443 | 28.68 | 64.70 | - | 0.413 | 24.49 | 59.54 | - |
| Spread-out | 0.440 | 24.54 | 57.16 | −1.18 | 0.479 | 32.12 | 66.83 | 3.44 | **0.458** | **31.85** | **67.45** | 7.36 |
| SVMax (Ours) | **0.527** | **41.26** | **75.24** | 15.53 | **0.547** | **43.11** | **77.26** | 14.43 | 0.449 | 29.56 | 65.50 | 5.06 |
| | | | | | | Triplet Loss | | | | | | |
| Vanilla | 0.496 | 29.34 | 67.96 | - | 0.477 | 28.88 | 64.60 | - | 0.449 | 24.86 | 61.14 | - |
| Spread-out | 0.545 | 43.60 | 76.98 | 14.26 | **0.557** | **44.02** | **78.54** | 15.14 | 0.435 | **28.33** | 64.33 | 3.46 |
| SVMax $\lambda = 1$ (Ours) | **0.556** | 43.21 | 77.43 | 13.88 | 0.527 | 39.13 | 74.17 | 10.25 | 0.401 | 25.07 | 60.01 | 0.20 |
| SVMax $\lambda = 0.1$ (Ours) | 0.547 | **43.80** | **77.97** | 14.47 | **0.557** | 43.89 | 78.44 | 15.01 | **0.436** | 28.22 | **64.40** | 3.36 |
| | | | | | | N-pair | | | | | | |
| Vanilla | 0.402 | 18.96 | 50.32 | - | 0.452 | 27.65 | 63.10 | - | 0.455 | 31.41 | 66.95 | - |
| Spread-out | 0.416 | 20.64 | 52.80 | 1.69 | 0.483 | 32.46 | 66.41 | 4.81 | 0.474 | 33.39 | 68.80 | 1.98 |
| SVMax (Ours) | **0.483** | **34.62** | **68.11** | 15.67 | **0.547** | **43.79** | **77.31** | 16.14 | **0.488** | **34.13** | **69.92** | 2.72 |
| | | | | | | Angular | | | | | | |
| Vanilla | 0.470 | 28.54 | 60.03 | - | 0.508 | 38.94 | 72.82 | - | **0.538** | 41.80 | 76.18 | - |
| Spread-out | .471 | 28.29 | 60.26 | −0.25 | 0.508 | 38.96 | 72.86 | 0.02 | **0.538** | 41.81 | 76.23 | 0.02 |
| SVMax (Ours) | **0.487** | **32.88** | **66.27** | 4.34 | **0.523** | **41.29** | **74.71** | 2.35 | 0.531 | **42.00** | **76.30** | 0.20 |

respectively, and their corresponding bounds $[L, U]$.

$$\text{TL}_{(a,p,n)\in T} = [(D(\lfloor a \rfloor, \lfloor p \rfloor) - D(\lfloor a \rfloor, \lfloor n \rfloor) + m)]_+ \xrightarrow{[L,U]} [0, 2 + m], \quad (5)$$

$$\text{CL}_{(x,y)\in P} = (1 - \delta_{x,y})D(\lfloor x \rfloor, \lfloor y \rfloor)) + \delta_{x,y}[m - D(\lfloor x \rfloor, \lfloor y \rfloor)]_+ \xrightarrow{[L,U]} [0, 2], \quad (6)$$

where $[\bullet]_+ = max(0, \bullet)$, $m < 2$ is the margin between classes, since 2 is the maximum distance on the unit circle. $\lfloor \bullet \rfloor$ and $D(,)$ are the embedding and Euclidean distance functions, respectively. In equation 5, $a$, $p$, and $n$ are the anchor, positive, and negative images in a single triplet $(a, p, n)$ from the triplets set $T$. In equation 6, $x$ and $y$ form a single pair of images from the pairs set $P$. $\delta_{x,y} = 1$ when $x$ and $y$ belong to different classes; zero otherwise. In the supplementary material, we (1) show similar analysis for N-pair and angular losses, (2) provide an SVMax evaluation on small training batches, $i.e.$, $b < d$, and (3) evaluate the computational complexity of SVMax.

## 4 EXPERIMENTS

In this section, we evaluate SVMax using both supervised and unsupervised learning. We leverage retrieval and generative adversarial networks for quantitative and qualitative evaluations.

### 4.1 RETRIEVAL NETWORKS

**Technical Details:** We evaluate the SVMax regularizer quantitatively using three datasets: CUB-200-2011 Wah et al. (2011), Stanford CARS196 Krause et al. (2013), and Stanford Online Products Oh Song et al. (2016). We use GoogLeNet Szegedy et al. (2015) and ResNet50 He et al. (2016); both pretrained on ImageNet Deng et al. (2009) and fine-tuned for $K$ iterations. These are standard retrieval datasets and architectures. By default, the embedding $\in R^{d=128}$ is normalized to the unit circle. In all experiments, a batch size $b = 144$ is employed, the learning rate $lr$ is fixed for $K/2$ iterations then decayed polynomially to $1e − 7$ at iteration $K$. We use the SGD optimizer with 0.9 momentum. Each batch contains $p$ different classes and $l$ different samples per class. For example, triplet loss employs $p = 24$ different classes and $l = 6$ instances per class. The mini-batch of N-pair loss contains 72 classes and a single positive pair per class, $i.e. p = 72$ and $l = 2$. This same mini-batch setting is used for angular loss. For contrastive loss, $p = 36$ and $l = 4$ are divided into 72 positive and 72 negative pairs. For CUB-200 and CARS196, $K = 5,000$ iterations; for Stanford Online Products, $K = 20,000$.

Table 2: Quantitative evaluation on Stanford Online Products.

| Method | lr = 0.01 | | | | lr = 0.001 | | | | lr = 0.0001 | | | |
|---|---|---|---|---|---|---|---|---|---|---|---|---|
| | NMI | R@1 | R@8 | $\triangle_{R@1}$ | NMI | R@1 | R@8 | $\triangle_{R@1}$ | NMI | R@1 | R@8 | $\triangle_{R@1}$ |
| | | | | | | Contrastive | | | | | | |
| Vanilla | 0.816 | 18.23 | 34.07 | - | 0.820 | 28.70 | 43.27 | - | 0.813 | 34.30 | 48.49 | - |
| Spread-out | 0.811 | 18.87 | 35.74 | 0.64 | 0.822 | 29.97 | 46.69 | 1.27 | 0.824 | 36.15 | 51.22 | 1.85 |
| SVMax (Ours) | **0.875** | **61.82** | **78.90** | 43.59 | **0.854** | **53.94** | **70.92** | 25.25 | **0.832** | **41.96** | **57.44** | 7.66 |
| | | | | | | Triplet Loss | | | | | | |
| Vanilla | **0.891** | **71.96** | **86.24** | - | **0.873** | 64.09 | 80.07 | - | **0.840** | 46.29 | 62.57 | - |
| Spread-out | 0.890 | 71.60 | 85.73 | −0.36 | 0.872 | **64.23** | 80.10 | 0.14 | **0.840** | **46.68** | **63.04** | 0.39 |
| SVMax $\lambda = 1$ (Ours) | 0.868 | 63.82 | 80.95 | −8.15 | 0.857 | 58.04 | 75.14 | −6.04 | 0.836 | 44.62 | 60.76 | −1.67 |
| SVMax $\lambda = 0.1$ (Ours) | 0.889 | 71.48 | 85.97 | −0.49 | 0.872 | **64.23** | **80.14** | 0.14 | **0.840** | 46.64 | 62.95 | 0.35 |
| | | | | | | N-pair | | | | | | |
| Vanilla | 0.798 | 12.86 | 24.53 | - | 0.815 | 23.83 | 38.97 | - | 0.818 | 33.98 | 48.56 | - |
| Spread-out | 0.803 | 16.58 | 31.91 | 3.72 | 0.824 | 32.88 | 50.34 | 9.05 | 0.825 | 37.39 | 52.55 | 3.40 |
| SVMax (Ours) | **0.871** | **57.76** | **76.05** | 44.90 | **0.858** | **54.70** | **71.57** | 30.87 | **0.835** | **43.04** | **58.78** | 9.06 |
| | | | | | | Angular | | | | | | |
| Vanilla | 0.883 | 62.83 | 80.13 | - | **0.885** | 66.93 | 82.12 | - | **0.856** | 54.29 | 71.14 | - |
| Spread-out | 0.883 | 62.73 | 79.96 | −0.10 | **0.885** | 66.91 | 82.09 | −0.02 | **0.856** | 54.30 | 71.10 | 0.02 |
| SVMax (Ours) | **0.885** | **65.44** | **81.73** | 2.61 | 0.884 | **67.28** | **82.47** | 0.35 | 0.855 | **54.88** | **71.47** | 0.59 |

**Baselines:** We evaluate the SVMax regularizer using contrastive Hadsell et al. (2006), hard triplet Hoffer & Ailon (2015); Hermans et al. (2017), N-pair Sohn (2016) and angular Wang et al. (2017) losses. We use the margin $m = 1$ for contrastive loss, $m = 0.2$ for triplet loss, and the angle bound $\alpha = 45°$ for angular loss. Similar to SVMax, multiple regularizers Kumar et al. (2016); Zhang et al. (2017); Sanyal et al. (2019); Chen & Deng (2019) promote a uniform embedding space. Unlike SVMax, these regularizers require a supervised setting to push anchor-negative pairs apart. We employ the spread-out regularizer Zhang et al. (2017) as a baseline for its simplicity, with default hyperparameter $\alpha = 1$. To enable the spread-out regularizer on non-triplet ranking losses, we pair every anchor with a random negative sample from the training mini-batch.

**Evaluation Metrics:** For quantitative evaluation, we use the Recall@K metric and Normalized Mutual Info (NMI) on the test split.

**The hyperparameter:** $\lambda = 1$ for both contrastive and N-pair losses, $\lambda = 0.1$ for triplet loss, and $\lambda = 2$ for angular loss. We fix $\lambda$ across datasets, architectures, and other hyperparameters $(b, d)$.

**Results:** Tables 1 and 2 present quantitative retrieval evaluation on CUB-200 and Stanford Online Products datasets – both using GoogLeNet. These tables provide in depth analysis and emphasize our improvement margins on a small and large dataset. Figure 2 provides quantitative evaluation on Stanford CARS196. We report the qualitative retrieval evaluation and quantitative evaluation on ResNet50 in the supplementary material. Our training hyperparameters – learning rate $lr$ and number of iterations $K$ – do not favor a particular ranking loss.

We evaluate SVMax on various learning rates. A large learning rate, *e.g.*, $lr = 0.01$, speeds up convergence, but increases the chance of model collapse. In contrast, a small rate, *e.g.*, $lr = 0.0001$, is likely to avoid model collapse but is slow to converge. This undesirable effect is tolerable for small datasets – where increasing the number of training iterations $K$ does not drastically increase the overall training time – but it is infeasible for large datasets. For contrastive and N-pair losses, SVMax is significantly superior to both the vanilla and spread-out baselines, especially with a large learning rate. A small $lr$ slows convergence and all approaches become equivalent. The spread-out regularizer Zhang et al. (2017) and its hyperparameters are tuned for triplet loss. Thus, for this particular ranking loss, the SVMax and spread-out regularizers are on par.

In our experiments, we employ a large learning rate because it is the simplest factor to introduce a model collapse. However, the learning rate is not the only factor. Another factor is the training dataset size and its intra-class variations. A small dataset with large intra-class variations increases the chances of a model collapse. For example, a pair of dissimilar birds from the same class justifies a model collapse when coupled with a large learning rate. The hard triplet loss experiments emphasize this point because every anchor is paired with the hardest positive and negative samples.

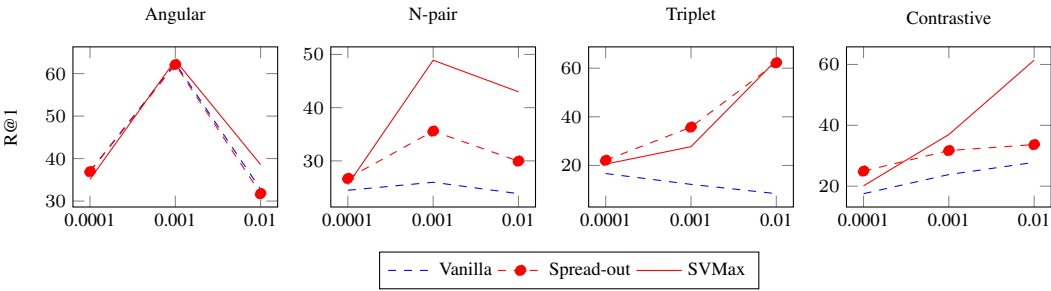

Figure 2: Quantitative evaluation on Stanford CARS196. X and Y-axis denote the learning rate $lr$ and recall@1 performance, respectively.

On small fine-grained datasets like CUB-200 or CARS196, the vanilla hard triplet loss suffers significantly. Yet, the same implementation is superior on a big dataset like Stanford Online Products. By carefully tuning the training hyperparameter on CUB-200, it is possible to avoid a degenerate solution. However, this tedious tuning process is unnecessary when using either the spread-out or the SVMax regularizer.

The vanilla N-pair loss underperforms because it does not support feature embedding on the unit circle. Both spread-out and SVMax mitigate this limitation. For angular loss, a bigger $\lambda = 2$ is employed to cope with the angular loss range. SVMax is a class oblivious regularizer. Thus, $\lambda$ should be significant enough to contribute to the loss function without dominating the ranking loss.

Wu et al. (2017) show that the distance between any anchor-negative pair, which is randomly sampled from an $n$-dimensional unit sphere, follows the normal distribution $N(\sqrt{2}, \frac{1}{2n})$. This mean distance $\sqrt{2}$ is large relative to the triplet loss margin $m = 0.2$, but comparable to the contrastive loss margin $m = 1$. Accordingly, triplet loss converges to zero after a few iterations, because most triplets satisfy the margin $m = 0.2$ constraint. When triplet loss equals zero, the SVMax regularizer with $\lambda = 1$ becomes the dominant term. However, the SVMax regularizer should not dominate because it is oblivious to data annotations; it equally pushes anchor-positive and anchor-negative pairs apart. Reducing $\lambda$ to $0.1$ solves this problem.

A less aggressive triplet loss Schroff et al. (2015); Xuan et al. (2020) is another way to avoid model collapse. For instance, Schroff et al. (2015) have proposed a triplet loss variant that employs semi-hard negatives. The semi-hard triplet loss is more stable than the aggressive hard triplet and lifted structured losses Oh Song et al. (2016). Unfortunately, the semi-hard triplet loss assumes a very large mini-batch ($b = 1,800$ in Schroff et al. (2015)), which is impractical. Furthermore, when model collapse is avoided, aggressive triplet loss variants achieve superior performance Hermans et al. (2017). In contrast, the SVMax regularizer only requires a larger mini-batch than the embedding dimension, i.e., $b \geq d$, a natural constraint for retrieval networks which favor compact embedding dimensions. Additionally, SVMax does not make any assumption about the sampling procedure. Thus, unlike Sablayrolles et al. (2018); Zhang et al. (2017), SVMax supports various supervised ranking losses.

## 4.2 GENERATIVE ADVERSARIAL NETWORKS

Model collapse is one of the main challenges of training generative adversarial networks (GANs) Metz et al. (2017); Srivastava et al. (2017); Mao et al. (2019); Salimans et al. (2016). To tackle this challenge, Metz et al. (2017) propose an unrolled-GAN to prevent the generator from overfitting to the discriminator. In an unrolled-GAN, the generator observes the discriminator for $l$ steps before updating the generator's parameters using the gradient from the final step. Alternatively, we leverage the simpler SVMax regularizer to avoid model collapse. We evaluate our regularizer using a simple GAN on a 2D mixture of 8 Gaussians arranged in a circle. This 2D baseline Metz et al. (2017); Srivastava et al. (2017); Bang & Shim (2018) provides a simple qualitative evaluation and demonstrates SVMax's potential in unsupervised learning. We leverage this simple baseline because we assume $b \geq d$, which does not hold for images.

| Method | Step 1 | Step 5k | Step 10k | Step 15k | Step 20k | Step 25k | Target |
|---|---|---|---|---|---|---|---|
| Vanilla GAN | | | | | | | |
| Vanilla GAN + SVMax | | | | | | | |
| Unrolled GAN (5 steps) | | | | | | | |
| Unrolled GAN (5 steps) + SVMax | | | | | | | |

Figure 3: The SVMax regularizer mitigates model collapse in a GAN trained on a toy 2D mixture of Gaussians dataset. Columns show heatmaps of the generator distributions at different training steps (iterations). The final column shows the groundtruth distribution. The first row shows the distributions generated by training a vanilla GAN suffering a model collapse. The second row shows the generated distribution when penalizing the generator's fake embedding with the SVMax regularizer. The third and fourth rows show two distributions generated using an unrolled-GAN with and without the SVMax regularizer, respectively. This high resolution figure is best viewed on a screen with zoom capabilities.

Figure 3 shows the dynamics of the GAN generator through time. We use a public PyTorch implementation[1] of Metz et al. (2017). We made a single modification to the code to use a relatively large learning rate, *i.e.*, $lr = 0.025$ for both the generator and discriminator. This single modification is a simple and fast way to induce model collapse. The mixture of Gaussians circle has a radius $r = 2$, *i.e.*, the generated fake embedding is neither L2-normalized nor strictly bounded by a network layer. We kept the radius parameter unchanged to emphasize that neither L2-normalization nor strict-bounds are required. To mitigate the impact of lurking variables (*e.g.*, random network initialization and mini-batch sampling), we fix the random generator's seed for all experiments. We apply SVMax to a vanilla and an unrolled GAN for five steps. We apply the vanilla SVMax regularizer (Eq. 1), *i.e.*, $L_{\mathrm{NN}} = L_{\mathrm{GAN}} - \lambda s_\mu$, where $\lambda = 0.01$ and $s_\mu$ is mean singular value of the generator fake embedding.

GANs are typically used to generate high resolution images. This high-resolution output is the main limitation of the SVMax regularizer. The *current* formulation assumes the batch size is bigger than the embedding dimension, *i.e.*, $b \geq d$. This constraint is trivial for the Gaussians mixture 2D dataset and retrieval networks with a compact embedding dimensionality (*e.g.*, $d = \{128, 256\}$). However, this constraint hinders high resolution image generators because the mini-batch size constraint becomes $b \geq W \times H \times C$, where $W$, $H$, and $C$ are the generated image's width, height, and number of channels, respectively. Nevertheless, this GAN experiment emphasizes the potential of the SVMax regularizer in unsupervised learning.

## 4.3 Ablation Study

In this section, we evaluate two hypotheses: (1) the SVMax regularizer boosts retrieval performance because it learns a uniform feature embedding, (2) the same SVMax hyperparameter $\lambda$ supports different embedding dimensions and batch sizes – the main objective of the mean singular value's bounds analysis.

To evaluate the SVMax regularizer's impact on feature embeddings, we embed the MNIST dataset onto the 2D unit circle. In this experiment, we used a tiny CNN (one convolutional layer and one hidden layer). Figure 4 shows the embedding space after training for $t$ epochs. When using the SVMax regularizer, the feature embeddings spread out more uniformly and rapidly than the vanilla contrastive loss.

---

[1]https://github.com/andrewliao11/unrolled-gans

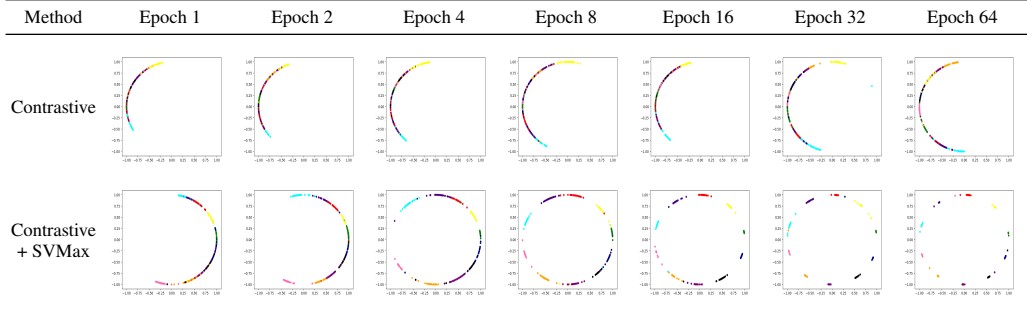

Figure 4: Qualitative feature embedding evaluation using the MNIST dataset projected onto the 2D unit circle. The first row shows the feature embedding learned using a vanilla contrastive loss and the second row applies the SVMax-regularized. A random subset of the test split is projected for visualization purpose. Different colors denote different classes. The regularized feature embedding spreads out uniformly and rapidly. The supplementary material shows the feature embedding evolves vividly up to 200 epochs. This high resolution figure is best seen on a screen.

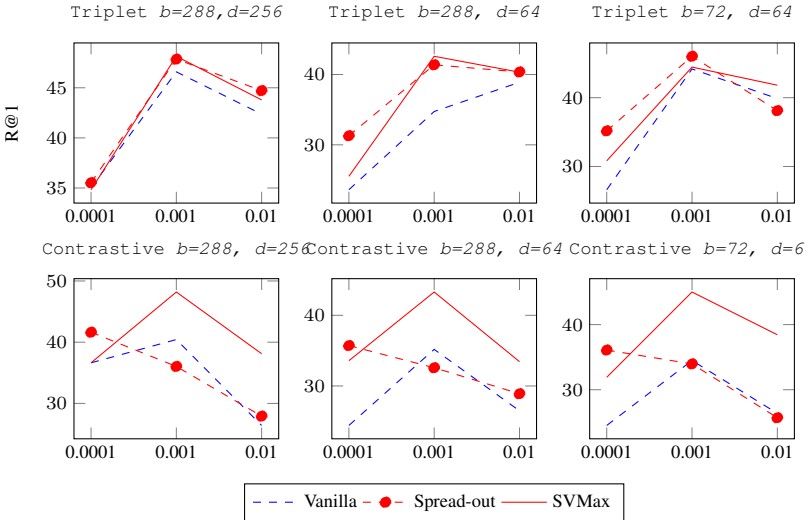

Figure 5: Quantitative evaluation on CUB-200-2011 with various batch sizes $b = \{288, 72\}$ and embedding dimensions $d = \{256, 64\}$ to demonstrate the stability of our hyperparameter. $\lambda = 1$ for contrastive loss and $\lambda = 0.1$ for triplet loss.

The mean singular value bound analysis makes tuning the hyperparameter $\lambda$ easier. This hyperparameter becomes only dependent on the ranking loss and independent of both the batch size and the embedding dimension. Figure 5 presents a quantitative evaluation using the CUB-200 dataset. We explore various batch sizes $b = \{288, 72\}$ and embedding dimensions $d = \{256, 64\}$. We employ a MobileNetV2 Sandler et al. (2018) to fit the big batch $b = 288$ on a 24GB GPU. The supplementary material contains a similar evaluation on the Stanford Online Products and CARS196 datasets.

## 5 CONCLUSION

We have proposed singular value maximization (SVMax) as a feature embedding regularizer. SVMax promotes a uniform embedding, mitigates model collapse, and enables large learning rates. Unlike other embedding regularizers, the SVMax regularizer supports a large spectrum of ranking losses. Moreover, it is oblivious to data annotation and, as such, supports both supervised and unsupervised learning. Qualitative evaluation using a generative adversarial network demonstrates SVMax's potential in unsupervised learning. Quantitative retrieval evaluation highlight significant performance improvements due to the SVMax regularizer.

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
