# OpenReview forum: "SVMax: A Feature Embedding Regularizer"
_ICLR.cc/2021/Conference — Reject_

### Official Review · AnonReviewer3 · 2020-10-28
**A simple regularizer, but the performance is yet to be validated**

**Rating:** 5
**Confidence:** 5

**Review:**

I have reviewed this paper in this year's Neurips. At that time, reviewers and AC have some good points but are not yet addressed (I have compared the two submissions carefully).

This paper proposes a Feature Embedding Regularizer (named SVMax) to regularize feature embedding during the CNN learning process. The idea is to push the distribution of high-dim feature embeddings to be a uniform distribution across the embedding space. The idea is implemented by adding a regularize to maximize the averaged singular value computed from a mini-batch. Experiments show that many deep metric learning methods can benefit from the proposed SVMax regularizer.

Pros:

1) A simple but effective regularizer.
2) This paper is well-written and easy to follow.

Cons:
I re-emphasize some major points pointed by reviewers and AC:
1) [From other reviewers] The experimental results seem to be worse than SOTA and the baseline method seems to be not well trained. For example, on CUB datasets, the baseline Trip method can reach above 50% R@1, but your result is only 47.7%. The same phenomenon happens on other baseline methods and backbone network ResNet50 (To my best knowledge, ResNet50 baseline can achieve 60%+ R@1.) So the experimental results make me confused about the effectiveness of the proposed method.

2) Though the effectiveness of the proposed SVMax Regularizer is validated on the fine-grain image retrieval datasets. I would also want to see its performance on a broader image retrieval task. For example:

Revisiting Oxford and Paris: Large-Scale Image Retrieval Benchmarking

The reason is that for this broader image retrieval task, the original feature embeddings may have been well-distributed across the embedding space, rather than shrunk to a limit embedding space for the fine-grain datasets.

I think the above experiment needs to be done if we want to draw the conclusion that SVMax regularizer really works for deep metric learning methods.

---

> ### Author Response · Authors · 2020-11-12
> **Addressed comments**
>
> Thank you for reviewing our paper. We appreciate that you mention many deep metric learning methods can benefit from our proposed SVMax regularizer.
>
> **Regarding experimental results.**  For fair comparisons, we did not tune our hyperparameters to a particular ranking loss. Thus, the 2.3% lag (50-47.7) is expected. SOTA results are generally achieved by tuning hyperparameters (lr, # iterations) for a particular ranking loss on a particular architecture. The best hyperparameters for angular loss achieve inferior performance on contrastive loss. The embedding dimension is another important hyperparameter. We opted to implement our baselines, choose learning rates uniformly [0.01, 0.001, 0.0001], and use default settings (e.g., margin) whenever possible.
>
> **Instance retrieval dataset.** The datasets present in our paper (CUB, Cars196, and Stanford Online Products) are the standard datasets in the community of metric learning and learning embeddings (please refer to [MLRC]). The Image Retrieval dataset recommended by you is an **instance retrieval** dataset, suited more for evaluating image descriptors, local geometric features, and geometric matching. To the best of our knowledge, no recent works on learning better embeddings evaluate on **instance retrieval** benchmarks.
>
> [MLRC] Musgrave, Belongie, Lim. 'A Metric Learning Reality Check'

---

### Official Review · AnonReviewer4 · 2020-10-28
**Good paper, novel idea, few missing points**

**Rating:** 6
**Confidence:** 3

**Review:**

Summary:

This paper proposes a new approach to regularize the feature embedding of neural networks. The proposed regularizer, maximizes the mean singular value of the feature matrix per batch, leading to a uniform spread of features. This enables learning with larger learning rates without the risk of model collapse. Authors derive lower and upper bounds for the proposed singular values loss, as well as popular ranking losses used in recent studies, eg. triplet loss and pairwise loss. These bounds help to tune the mixing parameter of network’s loss and the singular value loss.

Pros:

1- The paper considers a crucial problem of learning with neural networks, ie. model collapse. The approach is systematic and shows promising results for learning with large learning rates, while a model without regularization fails to perform well.

2- Authors derive lower and upper bounds for the proposed singular value loss, as well as well-known ranking losses in the literature.


Cons or comments:

1- Even though there is a guide to choose parameter $\lambda$ using the bounds on losses, authors use a fixed value in the experimental setup. In case that this value is according to the bounds for the particular setting, it would be nice that authors mention it.

2- The proposed regularizer is particularly useful with a larger learning rate. Then, one expects that less number of epochs would be enough to converge to a good state of network. It seems that the number of epochs in the retrieval experiments is the same across methods. If SVMax with lr=0.01 and spread-out with lr=0.0001 use the same number of epochs, there will be not much gain using SVMax.

3- Model collapse may appear in different modes, eg. unrolled GAN paper discusses these modes for GANs. It would be nice if authors investigate how the model collapse happens. For example in Table 1, lr=0.01, contrastive loss, vanilla and spread-out show very poor performance. One can investigate the principal components of the batch embeddings.

4- Authors did not mention how they exactly incorporate the mean singular value into the loss. Is the exact value computed for each mini-batch? Or an estimation would be considered? How much computation overhead does the mean singular value loss add?

---

> ### Author Response · Authors · 2020-11-12
> **Added clarifications**
>
> Thank you for taking the time to review our work. We are glad that you found our approach systematic and that it shows promising results.
>
> **Setting $\lambda$.**  We did attempt to address this in the paper on page 6. Specifically, we found that a larger $\lambda = 2$ was needed to cope with the larger angular loss range. Additionally, when triplet loss is zero $\lambda = 1$ dominates training and we found that reducing $\lambda$ to $0.1$ solves this issue.
>
> **Comparing results between learning rates.**  Our experiments show that your claim is not consistently true. For instance, we significantly outperform spread-out on Contrastive and N-Pair losses in Table 2. As we stated in the paper, larger learning rates tend to induce model collapse. While SVMax appears to mitigate that effect, we suspect that this is the reason for the slight drop in accuracy as we increase learning rate. However, our results seem to show that we are much more robust to this drop than existing methods.
>
> **Baseline principal components.**  t-SNE visualizations did not provide clear insights, but we’ll explore the reason behind lower baseline performance further.
>
> **Singular value computation.**  This information is not in the main paper, but we did include it in section B2 of our appendix (pages 6 and 7). Specifically, we use the TensorFlow built-in function `tf.linalg.svd` to compute the mean singular value. We calculate this for each mini-batch. We did not notice a significant overhead. For more details see Figure A10.

---

### Official Review · AnonReviewer2 · 2020-10-29
**Borderline Accept**

**Rating:** 6
**Confidence:** 5

**Review:**

Pros:
This paper is well written and easy to follow. It is also well organized from intuition to method and then to experiments. The background of the proposed method is interesting and useful. By regularizing the distribution of learned features is more likely to be benefit for robust embedding learning.The proposed method SVMax is novel to me. It has rigorous mathematical guarantees and analysis.Extensive experiments are conducted to demonstrate what the authors claimed. And corresponding analysis are also provided.

Cons:
It seems that the proposed method cannot consistently improve the performances over the baseline method, such as Angular-Loss et.al. What is the reason in fact?

---

> ### Author Response · Authors · 2020-11-12
> **Added clarifications**
>
> We appreciate that you’ve taken the time to review our work and are glad you found it easy to follow and that we’ve been extensive in our experiments.
>
> **Angular loss performance.**  We’d like to highlight that, when our approach is lower than Angular loss, the margins are quite small (e.g., 0.001 in Table 2 and 0.007 in Table 1). Therefore, we posit that under those hyperparameter setups, both approaches exhibit similar behavior.

---

### Official Review · AnonReviewer1 · 2020-10-29
**Promising ideas, but insufficient motivation, analysis, and comparison**

**Rating:** 4
**Confidence:** 3

**Review:**

This paper proposes a regularization technique called SVMax (singular value maximization) that can mitigate model collapse and enable large learning rates to reduce training computation costs. The singular value decomposition of network activation is used to regularize the embedding space with unit circle embedding assumptions. In addition, a mathematical analysis of the mean singular value boundary is provided to reduce hyperparameter tuning. The authors evaluate the proposed method for the retrieval tasks and generative adversarial networks.

The idea looks promising in that it is simple and easy to use. However, the paper is a bit confusing and lacks clarity. I hope that the motivation for the introduction of the algorithm, analysis, and comparison experiments will be revealed well with care. A major revision of the work is needed.

Pros:
+ By introducing a simple method, it leads to good performance within the experimental work.
+ The flow of converting the former regularizer expression by mathematically analyzing the lower and upper bounds of the mean singular value is clear and well.
+ It is impressive that the corner cases of the proposed algorithm are written in the supplementary material. I agree that the corner case will not really happen.

Cons:
- Insufficient motivation. It seems not enough to simply apply singular value decomposition to the regularizer. In addition to the good results of the experiment, it is necessary to add more mathematical analysis, proof, etc. to see what this regularizer makes good.
- Is it best for model collapse mitigation or large learning rate activation to be seen only as experimental results? Hopefully, there is something in this part that can analyze why SVMax is possible.
- Why maximize mean singular value? Is maximizing the mean singular value the only way to spread out deep network embeddings? Can't maximize the min or max of singular values? It would be great if any comparison or analysis of this was supplemented.
- I'm confused about how the lower and upper bounds of loss functions are used and why they are in the paper.
- I would like to have various experimental results. In addition to the retrieval task, it would be nice if there was an experiment result from a classification task. Or, for example, it is better to experiment with local patch descriptors evaluated in your baseline algorithm paper.

---

> ### Author Response · Authors · 2020-11-12
> **Addressed comments; Added clarifications**
>
> Thank you for taking the time to review our work. We took particular care to analyze the mean singular value bounds and appreciate that you recognize this as our strength.
>
> **Mathematical analysis / SVMax analysis.**  Can you be more specific about what mathematical analysis and proofs are missing and will add to the exposition?  We believe that we’ve made a strong case for our regularizer and have provided adequate theory, intuition, and empirical results. But we will happily oblige if there are glaring omissions.
>
> **Other methods to spread embeddings.**  Certainly, developing other methods to spread out the embedding space is a fruitful research avenue. In this work, we argue that maximizing the mean singular value is a simple and effective way to accomplish this. We believe that by bringing attention to this problem and by proposing a simple and effective solution provides a benefit to the community. To your particular suggestion of maximizing the maximum singular value: this would, in fact, condense the embedding space and not spread it out.
>
> **Loss bounds.**  Coupling the bounds on the ranking losses with Equation 4 allows us to interpret the balancing hyperparameter, $\lambda$, as the proportional trade off between the ranking loss and the regularizer. Additionally, these loss bounds show that the loss will never get ‘too big’ and dominate the regularizer.
>
> **Other tasks.**  We agree that additional tasks would further support SVMax. But given that we’ve shown improvements on both retrieval and model collapse, we believe that SVMax will be of benefit to the community and should not be dismissed for simply not evaluating it on more tasks. Regarding classification: it’s unclear how more uniform features would improve classification performance. We believe SVMax is best suited for use in feature embedding settings.

---

### Decision · Program_Chairs · 2021-01-07
**Final Decision**

**Decision:**

Reject

**Comment:**

The paper presents a new regularizer based on singular value decomposition in embedding space to avoid model collapse. The reviewes liked the simplicity of the idea, but there were some remaining concerns regarding the experiments. Moreover, two reviewers mentionned some concerns with respect to the clarity of the paper. While some concerns have been addressed by the rebuttal, in particular regarding the clarity of the paper, the concerns regarding the experiments remained, and the reviewers agreed that the paper needs a revision before publication.

The main directions of improvement are to make the comparison with previous published results clearer, in particular comparing different methods with better hyperparameter tuning, and test on larger datasets.